# Novel Exon 7 Deletions in *TSPAN12* in a Three-Generation FEVR Family: A Case Report and Literature Review

**DOI:** 10.3390/genes14030587

**Published:** 2023-02-25

**Authors:** Zixuan Jiang, Panfeng Wang

**Affiliations:** State Key Laboratory of Ophthalmology, Zhongshan Ophthalmic Center, Sun Yat-sen University, Guangzhou 510060, China

**Keywords:** FEVR, *TSPAN12*, novel phenotypes, CNV, intrafamilial variability

## Abstract

Familial exudative vitreoretinopathy (FEVR) is a severe clinically and genetically heterogeneous disease that is characterized by vascular disorder. FEVR exhibits strikingly variable clinical phenotypes, ranging from asymptomatic to total blindness. In this case, we present a patient who was first treated as having high myopia and retinopathy but was finally diagnosed with FEVR caused by the heterozygous deletion of exon 7 in *TSPAN12* with the aid of whole genome sequencing (WGS). Typical vascular changes, including vascular leakage and an avascular zone in the peripheral retina, were observed in the proband using fundus fluorescein angiography (FFA), and the macular dragging was shown to be progressing in the follow-up visit. Furthermore, the proband showed unreported *TSPAN12*-related phenotypes of FEVR: ERG (full-field electroretinogram) abnormalities and retinoschisis. Only mild vascular changes were exhibited in the FFA for the other three family members who carried the same deletion of exon 7 in *TSPAN12*. This case expands our understanding of the phenotype resulting from *TSPAN12* mutations and signifies the importance of combining both clinical and molecular analysis approaches to establish a complete diagnosis.

## 1. Introduction

Familial exudative vitreoretinopathy (FEVR), with an incidence of 0.11% among newborns [1], is a rare inherited vitreoretinopathy characterized by incomplete vascularization and poor differentiation in the peripheral retina [2]. The mean age of onset is 6 years, although it can present at any age. FEVR is clinically and genetically heterogeneous, and it has three inherited forms: autosomal dominant, autosomal recessive, and X-linked recessive. Thus far, 11 genes have been implicated in the pathogenesis of FEVR, namely *NDP, FZD4, LRP5, TSPAN12, ZNF408, KIF11, RCBTB1, CTNNB1, ILK, JAG1,* and *ATOH7* [3]. Among these, *NDP, FZD4, LRP5, TSPAN12*, and *CTNNB1* have been proven to participate in the Norrin/β-catenin signaling pathway, which plays a vital role in retinal angiogenesis [4]. The symptoms of FEVR vary widely among different patients and even patients in the same family [5]. Mild FEVR may be asymptomatic, manifesting only as peripheral retinal vascular abnormalities. The clinical manifestations of severe FEVR include retinal neovascularization, subretinal and intraretinal hemorrhages, exudates, retinal folds, and retinal detachment [6,7,8].

Mutations in *TSPAN12* account for 5.6% to 8.0% of FEVR patients and are more common (12.8%) in patients with asymptomatic mild FEVR [9,10,11]. Currently, there are 103 mutations marked as DMs (disease-causing mutations) in the HGMD (Human Gene Mutation Database) (2022.1), and only 8 of them are CNV (copy number variant) changes. In this study, we discovered a family affected by FEVR due to a copy number variant covering exon 7 deletion in *TSPAN12* (NM_012338.4).

## 2. Materials and Methods

### 2.1. Clinical Evaluation

The FEVR proband was recruited from the Pediatric and Genetic Clinic, Zhongshan Ophthalmic Center, Guangzhou, China. This study was approved by the institutional review board of the Zhongshan Ophthalmic Center. Written informed consent consistent with the tenets of the Declaration of Helsinki was acquired from the proband and his family members before the clinical records and venous blood samples were collected. Genomic DNA was prepared from leukocytes obtained from the venous blood using the Maxwell^®^16 DNA Purification kits (Promega (Beijing) Biotech Co., Ltd., Beijing, China) according to the manufacturer’s recommendations.

Ophthalmologic examinations, including fundus photographs, best corrected visual acuity (BCVA), scanning laser ophthalmoscopy (SLO), fundus fluorescein angiography (FFA), optical coherence tomography (OCT) (ReVue software version 2017.1.0.155, Optovue Inc., Fremont, CA, USA), and a full-field electroretinogram (ERG) were systemically conducted on the proband and available family members. ERG was recorded for both eyes with the Diagnosys Espion E2 ERG system (Diagnosys LLC, Lowell, MA, USA); the recordings fully complied with the ISCEV ERG standard, namely using corneal electrodes, mydriasis, and a standard DA sequence [12]. The angles between the temporal inferior and temporal superior venous arcades and the optic disks were measured using Adobe Photoshop software.

### 2.2. Targeted Exome Sequencing of Genes for Inherited Eye Diseases and Sanger Sequencing

Targeted exome sequencing and variant detection by multi-step bioinformatic analyses were performed as described in our previous study [13,14]. In brief, genomic DNA (gDNA) was fragmented into approximately 200 base-pair (bp) fragments with a Bioruptor Plus (Diagenode, Liege, Belgium), and then used to generate a paired-end library with a KAPA HTP Library Preparation kit (Roche, Basel, Switzerland). Library capture was completed using a NimbleGen SeqCap EZ Choice Library SR V5 kit (Roche), and the library was sequenced with an Illumina NextSeq 550 Mild output v2 kit (150 PE) on an Illumina Nextseq 550 analyzer (Illumina, San Diego, CA, USA). The average depth of the target region was 250×. Sanger sequencing was used to verify the point mutations. The sequences of the forward and reverse primers used to verify suspected variants were *CACNA1F*-F 5′-ACGCTGGATGAGATGGACAA-′3, *CACNA1F*-R 5′-GGGAGGGCAGGAGGTTTATT-′3, *CNGB1*-F 5′-TCATGACACTTCTCCCTGGG-′3, and *CNGB1*-R 5′-GGTTGAGCCCACTGGATTTA-′3.

### 2.3. Whole-Genome Sequencing (WGS), CNV Analysis, and Fluorescent Quantitative PCR (qPCR)

A NadPrep DNA Universal Library Preparation kit (for MGI) (Nanodigmbio, Nanjing, China) was used for library construction. The standard WGS library construction method for MGISEQ-2000RS includes the following major steps: (1) DNA fragmentation; (2) end repair and A-tailing; (3) indexed adapter ligation; (4) post-ligation cleanup; (5) library amplification; (6) post-amplification cleanup; and (7) DNB preparation. The libraries were then pooled, and a number of DNBs were generated; the products were measured using Qubit with the use of an ssDNA kit and were finally sequenced using an MGISEQ-2000RS (MGI Tech, Shenzhen, China), with an average depth of 30× [15]. Bioinformatic analyses were performed using both public software and software developed in-house. Specifically, fastp 0.20.0 quality control software was used to remove low-quality adaptors [16]. Read alignment was performed using the human genome assembly hg19 (GRCh37) and the MEM algorithm of Burrows-Wheeler Alignment 0.7.17 software [17]. The detection of SNVs and indels (<50 bp) was processed using GATK3.8 software [18]. Next, Variant Effect Predictor v100 software and corresponding plugins, as well as disease databases, such as OMIM (Online Mendelian Inheritance in Man) and ClinVar, were employed to annotate variants with population frequency (GnomAd) [19]. Copy number variants were uncovered using CNVkit 0.9.7, Delly v0.8.1 software was used for anomaly discovery, and annotation was performed using AnnotSV 2.5.0 [20,21,22]. The alignment data were divided into consecutive equal-length bins using CNVkit v0.9.7 software, and the depth and coverage of the bins were homogenized. Bins belonging to the training sample set under the same experimental and sequencing conditions were compared to obtain the abnormal copy number variant regions; then, the abnormal regions were cross-annotated with databases, including OMIM, ClinVar, and ClinGen, using AnnotSV 2.5.0 software. The Splice Site Prediction (https://www.fruitfly.org/seq_tools/splice.html, accessed on 12 January 2023) tool was used to predict whether the deletion would result in new splice sites. Relative copy number was measured via a quantitative real-time polymerase chain reaction (qPCR) assay using Applied Biosystems StepOnePlus (Applied Biosystems, Waltham, MA, USA). The *ALB* (Albumin) gene was used as a reference gene. Real-time PCR was performed on 20 μL of the reaction mixture, including 2X SYBR^®^ Green PCR Master Mix (A25742; Thermo Fisher Scientific Inc., Waltham, MA, USA), 10 μM forward primers, 10 μM reverse primers, 6 μL double-distilled water, and 8 ng DNA. A normal control (NC) was included in each run. The thermal profile included an initial activation step at 95 °C for 10 min, followed by 40 cycles at 95 °C for 15 s (the denaturation step) and 60° for 10 min (annealing/extension). For the amplification of a portion of exon 7 of *TSPAN12*, the following pair of primers were used: a forward primer, 5′-CCTGCTGTGTTAGAGAATTCCC -′3, and reverse primer, 5′-GGTCACTGAGATCTTCCTGGT -′3. The sequences of the forward and reverse primers used as a normal control were *ALB*-F 5′- TGAAACATACGTTCCCAAAGAGTTT-′3 and *ALB*-R 5′- CTCTCCTTCTCAGAAAGTGTGCATAT-′3, respectively. For each patient sample, PCR amplification was performed in triplicate. The copy number of exon 7 of the *TSPAN12* gene in the normal control (NC) was set as 2. Afterward, the normalized copy number was determined using the following equation [23]:ΔCT_NC = CT Mean (NC) − CT Mean (ALB)
ΔCT_sample = CT Mean (patient) − CT Mean (ALB)
ΔΔCT_sample = ΔCT_NC − ΔCT_sample
Copy number_sample = 2^ΔΔCT_sample^ × copy number_NC

Patients with copy numbers between 0.5 and 1.4 were determined to have a heterozygous deletion of exon 7, while patients with copy numbers between 1.5 and 2.4 were considered to have no deletion.

### 2.4. Reverse Transcription PCR (RT-PCR) and Sequence Analysis

Total RNA was prepared from leukocytes obtained from venous blood using the Maxwell^®^16 RSC simplyRNA Blood kit (Promega (Beijing) Biotech Co., Ltd., Beijing, China) according to the manufacturer’s recommendations. Then, complementary DNA (cDNA) was synthesized from an equal amount of total RNA (1 μg) according to the instructions in the PrimeScript^TM^ RT reagent kit with gDNA Eraser (Takara, Dalian, China). The cDNA was used to confirm the effect of deletion on the RNA level of *TSPAN12* through the gel view of PCR amplification as well as Sanger sequencing. To avoid gDNA amplification, a piece of F primer spanning the junction of exons 5 and 6 was designed: *TSPAN12*-E5/6-F1 5′- AGGAACTTATGGTTCCAGTACAA-′3, and a piece of R primer located in exon 8 was designed: *TSPAN12*-E8-R1 5′- TGCCATGGATGTGTTCAA-′3. The designed normal length of the PCR product was 530 bp and was verified using Primer-BLAST by NCBI (https://www.ncbi.nlm.nih.gov/tools/primer-blast/index.cgi), accessed on 12 January 2023. The sequences of the product were assembled using SnapGene software (www.snapgene.com, accessed on 12 January 2023).

### 2.5. 3D Structure Modeling of TSPAN12 Protein after Gross Deletion

The 3D structure of the TSPAN12 protein after the deletion of different regions was predicted by Phyre2 (http://www.sbg.bio.ic.ac.uk/phyre2, accessed on 16 February 2023). Advanced remote homology detection methods were performed to build 3D models, predict ligand binding sites, and analyze the effect of amino acid variants on protein sequence [24]. Model quality was assessed by ProQ2 [25]. Topology in the membrane and transmembrane helices was predicted by PSI-BLAST (accessed on 16 February 2023) [26]. The predicted 3D structure was visualized using Swiss PDB viewer 4.1 software (Swiss Institute of Bioinformatics, Lausanne, Switzerland).

## 3. Results

### 3.1. Patient Characteristics and Genetic Analysis

A two-and-a-half-year-old boy visited our clinic due to poor sight. He was born at full-term by cesarean section. His refraction was −0.0 diopters (D) in both eyes, and his best corrected visual acuity (BCVA) was not available because of his young age. Neither of his parents wore glasses; the father’s best vision was 1.0 in both eyes, while the mother’s refraction was around −2.5 D in both eyes, with a best vision of 1.0. A family history of high myopia was not reported by other available family members. Fundus examination revealed a pale optic disc. The ERG examination showed abnormal cone and rod cell mediated responses in both eyes, compared to the range of the mean values for this instrument in our hospital from healthy people with no significant ocular abnormalities and refractive errors. In the scotopic ERG, the amplitude of the b-wave of the rod response was 56.34 μV/46.92 μV (OD/OS) (reduced by 70.19% and 75.17%, respectively; severely reduced). The implicit time of the mixed response b-wave was 78 ms/83 ms (delayed by 36.84% and 31.33%, respectively; moderately delayed), and the amplitude of the b-wave was 241.6 μV/180.7 μV (reduced by 41.78% and 56.46%, respectively; moderately reduced). In the photopic ERG, the amplitude of the a-wave of the cone response was −31.56 μV/−30.79 μV (declined by 163.12% and 161.58%, respectively; severely reduced), while the amplitude of the b-wave of the cone response was 25.37 μV/31.1 μV (declined by 76.72% and 71.47%, respectively; severely reduced). Furthermore, the 30 Hz flicker response was severely reduced. There was no electronegative ERG (Figure 1a). According to the value of the reported standard (full-field) ERG, including emmetropia, myopia (mild, moderate, and high), and pathological myopia [27], the ERG abnormalities of this patient were highly similar to the pattern of pathological myopia. At the time of the initial visit, the patient was diagnosed with early onset high myopia (eoHM) based on the age of onset, refractive error, and the result of the fundus photograph appearance. It has been reported that early onset high myopia is strongly associated with systemic and ocular problems, and it may be the reason for the child’s initial medical referral [28]. These systemic syndromes can be caused by a series of genes, including *NYX*, *CACNA1F*, *GRM6*, and *LRIT3*, responsible for congenital stationary night blindness (CSNB) [29,30,31,32]; *COL2A1*, *COL11A1*, *COL9A1*, and *COL9A2*, responsible for Stickler syndrome [33,34,35,36]; and *FBN1*, responsible for Marfan syndrome [37,38]. Thus, a targeted exome sequencing of genes for inherited eye diseases was performed on genomic DNA obtained from the patient’s peripheral blood, and a hemizygous c.5881G>T (p.Ala1961Ser) variation in the *CACNA1F* gene and a homozygous c.1631C>T (p.Pro544Leu) variation in the *CNGB1* gene were found. For variant c.5881G>T (p.Ala1961Ser) in *CACNA1F*, it was observed in the gnomAD database with a frequency of 0.000652 in East Asian individuals. The bioinformatic analysis showed that the effect of this missense mutation was benign (REVEL score: 0.2829). However, in II:3 (the uncle of the proband), harboring the same hemizygous *CACNA1F* c.5881G>T variant, (Figure 2a,b), a normal ERG was detected, which did not support the co-segregation of this variant of *CACNA1F* in this family. For the homozygous variant, the c.1631C>T (p.Pro544Leu) variation in *CNGB1*, it was observed in the gnomAD database with a frequency of 0.00223 in East Asian individuals. The bioinformatic analysis showed that the effect of this missense mutation was also benign (REVEL score: 0.123). These two variations detected in early onset high myopia causative genes were classified as benign according to the American College of Medical Genetics and Genomics (ACMG) guidelines, and could not explain the phenotype of the proband [39]. This result led to the decision to perform whole-genome sequencing (WGS).

Further WGS detected a large deletion in the region of Chr7:g.120437365-g.120450365 (NM_012338.4: c.468+152_c. 613-8414del), which contained exon 7 of the *TSPAN12* gene (Figure 2c). The CNV was predicted to be likely pathogenic (LP) for two reasons: (1) a truncation was one of the *TSPAN12* gene’s causative mechanisms of FEVR, and (2) the CNV was missing from the DGV (Database of Genomic Variants) and 1000 genome database populations. A co-segregation analysis was performed using qPCR on available family members, and the heterozygous CNV was detected in the proband, I:1, II:2, and II:3 (Figure 2d). The effect of the CNV was predicted by bioinformatic analyses. A new splice site was ruled out, and an in-frame deletion of exon 7 was proved by RNA analyses (Figure 2e,f). In the normal RNA of *TSPAN12*, a fragment of 530 bp could be observed in agarose gel electrophoresis (Figure 2e). A shorter fragment (386 bp) appeared in the amplicon on the temple from the cDNA of II:2, which indicated a deletion happened in this amplification area. Sanger sequencing on the shorter fragment proved that the deletion was exon 7 of *TSPAN12* (Figure 2f). None of the pathogenic mutations in genes related to congenital retinoschisis (*RS1*) or retinal degeneration (*USH2A*, *ABCA4*, *PDE6A*, *PDE6B*, *RPE65*, etc.) were detected by targeted exome sequencing or WGS [40,41]. The 3D structure modeling analyses showed that the deletions of Exons1-3, 4, and 8 of TSPAN12 were predicted to destroy the ECL-1 or ECL-2 region separately. The ECL-2 of TSPAN12 with E7 deletion was predicted to be shorter (116-173 amino acids) than normal TSPAN12 (116-220 amino acids) (Figure 3).

Based on the results of the WGS, further clinical examinations of the patient and his family members were carried out, including FFA, SLO, or OCT. The patient’s FFA showed significant brush-like changes in the peripheral vessels of both eyes, especially on the temporal side. An avascular area was also observed on the temporal side, and there was vascular leakage in the temporal periphery (Figure 1b). The fluorescence imaging of I:1 and II:2 (Figure 1c,d, and Appendix A) also showed mild FEVR-related vascular changes. Three individuals with heterozygous CNV in the co-segregation analysis (the proband, I:1, II:2, and II:3) had abnormal FFA results (Figure 2a,d). FFA was not available for II:3 because of his allergy to fluorescein sodium. The OCT of the fundus showed that temporal retinoschisis between ILM and OPL was observed in both eyes of the proband (Figure 1e). The CNV change, Chr7:g.120437365–g.120450365, detected in this family was thus classified as LP (PVS1, PM2, PP1, and PP4).

In brief, the young patient was diagnosed with eoHM at the first visit because of refractive error. OCT and ERG examinations suggested that the patient had retinoschisis and ERG abnormalities. The diagnosis at this time favored a syndrome with early onset high myopia. Targeted sequencing was performed, revealing two variants in *CACNA1F* and *CNGB1* separately. However, both bioinformatic predictions and family co-segregation concluded that these two variants were benign. Whole-genome sequencing revealed a large copy number variation (*TSPAN12* c. 468+152_c. 613-8414del) that spanned the entire exon 7 of *TSPAN12*. Bioinformatic analysis and further analysis at the transcription level supported the fact that this CNV produced an in-frame deletion of the whole of exon 7 in *TSPAN12*. The FFA images and qPCR results of this family validated the co-segregation. According to the clinical and genetic results, the diagnosis of the proband changed from early onset high myopia to FEVR caused by the heterozygous deletion of exon 7 in *TSPAN12*.

### 3.2. Mild FEVR and Follow-Up Visits

Previously published literature showed that among asymptomatic mild FEVR patients, although only 1.61% (2 out of 124 eyes) would develop into severe situations that required laser photocoagulation treatment, all patients with mild FEVR need lifelong monitoring [9]. Follow-up examinations were performed on the proband, beginning with his first visit to our hospital and continuing every six months. We tracked the retina and vascular changes in the proband using FFA. The angles between the arcade vessels and centering on the optic disks changed from 110.1° to 99.7° in the right eye and from 107.1° to 97.4° in the left eye (Figure 1b). The smaller angle revealed that the macular dragging was progressing (Figure 1b); a similar situation was also observed in young patients whose macular dragging was reversed by retinal photocoagulations [42]. Changes in the degree of myopia of the proband between the ages of one year and nine months and four years and three months were recorded (Appendix A). Developing retinal vascular traction and enlarging retinoschisis were observed, but further surgery or laser treatment measures were not taken by now considering his age and macular non-involvement after multi-disciplinary treatment.

Interestingly, none of these three generations of family members manifested FEVR except for the proband. His grandfather and mother were asymptomatic but had fundus changes detected by FFA. Moreover, individuals with the same CNV in this family experienced different severities of vascular leakage, as shown by FFA. Thus, it was necessary to perform an FFA examination on carriers in the FEVR family [43].

## 4. Discussion

In this study, a heterozygous gross deletion on chromosome 7 between g.120437365 and g.120450365 was discovered in the FEVR family. Bioinformatic analyses predicted an in-frame deletion of the entire exon 7 in *TSPAN12* instead of splicing site changes, which was further confirmed by RT-PCR. *TSPAN12* is a member of the tetraspanin family that encodes for a 305-amino-acid transmembrane protein, including 4 transmembrane domains, a conserved CCG motif, and 2 other cysteine residues [44]. These four transmembrane domains are connected by two extracellular loops (ECL-1 and ECL-2) and an intracellular loop [45]. The CCG motif and cysteine residues contribute to two crucial disulfide bonds within the second extracellular loop (ECL-2), and they are crucial for forming disulfide bonds and protein folding. The ECL-2 domain is made up of the whole exon 7 and part of exon 8 and contains almost all of the known tetraspanin protein–protein interaction sites [46]. This functional study showed that TSPAN12 was anchored to the Norrin receptor complex via an interaction of the large extracellular loop with FZD4, and the ECL-2 domain of TSPAN12 was essential for enhancing Norrin-induced FZD4 signaling [47,48]. Furthermore, 38% of *TSPAN12* mutations identified in FEVR patients were concentrated in the ECL-2 domain [45]. These findings support the role of ECL-2 in the pathogenesis of FEVR.

Eight gross deletions in *TSPAN12* were detected in eight reported cases and four patients in this study, with the age of onset ranging from 0.2 years to 52 years (Figure 2g, Appendix A). Four of them harbored whole-gene deletion, and most of their eyes (87.5%, 7/8) were classified as stage 1 or 2 [2,10]; another four harbored partial exon deletion (exon 1–3, 4, and 8), and most of their eyes (75%, 6/8) were classified as stage 3 or 4 [2,10,49,50]. Interestingly, the remaining four harbored exon 7 deletions detected by this study, where all eight available eyes were classified as stage 1 or 2 (Appendix A), according to the Kashani classification system [51]. Although the exon 7 and exon 8 were located in the same ECL-2 region, the reported patient with an exon 8 deletion presented with the tractional detachment of the retina [50], while all four individuals in this study with an exon 7 deletion had mild FEVR, indicating a peripheral avascular zone in the retina. To reveal the possible mechanism, we constructed 3D protein models of TSPAN12 with all these reported types of gross deletions. Exons 1–3, 4, and 8 were predicted to destroy ECL-1 and ECL-2 separately (Figure 3), whereas exon 7 was predicted to shrink ECL-2. It appears that the severity of FEVR caused by CNV changes in *TSPAN12* is dependent on the location of the deletion and is not related to the size of the deletion or the age of onset. The extracellular loops are important to the normal function of TSPAN12, and the destruction of these areas would result in more severe FEVR. The relationship between the genotype and the phenotype should be further confirmed in more large-scale studies.

Patients with FEVR caused by *TSPAN12* usually have abnormal retinal vasculogenesis, leading to peripheral retinal non-perfusion, ischemia, fibrovascular proliferation, and retinal detachment [52]. In addition to these common phenotypes, some atypical features were also reported in our previous study, including glaucoma and retinitis pigmentosa [53]. However, ERG analysis and investigation of the retinal structure were rarely carried out. In this study, the proband presented ERG abnormalities and retinoschisis. Because the proband also had high myopia, and his ERG values were highly similar to pathological myopia mean ERG values, the relationship between *TSPAN12* gene mutations and ERG abnormalities requires further investigation. In addition, since genes known to be associated with retinoschisis were ruled out by WGS in the proband, retinoschisis might have unreported phenotypes caused by the *TSPAN12* mutation; however, more clinical cases are needed to confirm these.

The diagnosis of FEVR has remained difficult due to its clinical and genetic heterogeneities. Moreover, the vascular changes in early stage FEVR are subtle; thus, clinical examination alone can be insufficient to diagnose FEVR, and imaging findings of FFA are required [54]. They are essential for the diagnosis and staging of this disease, and they can also identify the disease in its earlier stages to facilitate treatment in a timely manner and help identify asymptomatic family members who may be affected [55].

In this report, we present a young boy who was initially diagnosed with early onset high myopia due to nearsightedness since childhood. Through two genetic testing methods combined with imaging findings, the diagnosis of FEVR was finally established. In addition, we found three family members carrying the same deletion through the qPCR analysis. Their vascular abnormality could only be detected by FFA. While the symptoms of mild FEVR are slight and easy to ignore, this report suggests that WGS can be helpful in the diagnosis of this kind of disease. Our findings highlight the importance of paying attention to the peripheral retina during fundus examination in patients with early onset high myopia and further affirm the usefulness of next-generation sequencing to guide the early diagnosis and symptomatic management of inherited eye diseases such as FEVR.

## Figures and Tables

**Figure 1 genes-14-00587-f001:**
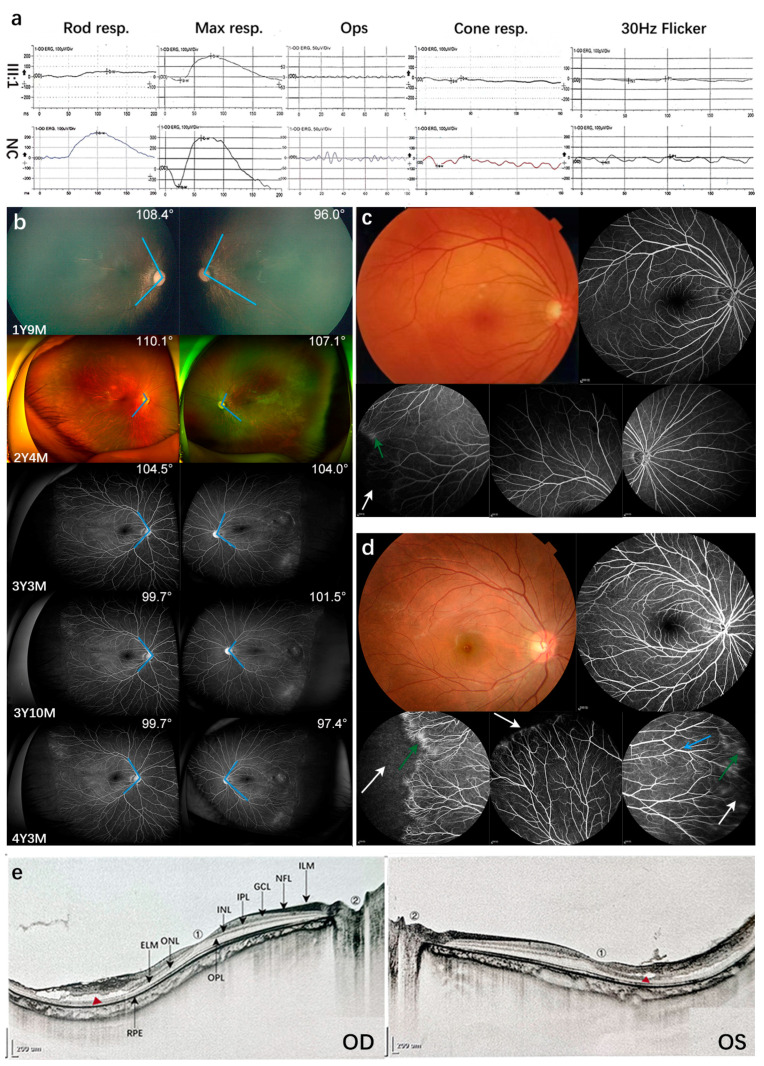
Ocular phenotypes observed in the family. (**a**) Abnormalities of ERG were recorded in the left eye of III:1 and compared with the normal control (NC). NC had no significant ocular abnormalities or refractive errors. (**b**) Typical vascular changes, including vascular leakage and an avascular zone in the peripheral retina, were observed in the proband using fundus fluorescein angiography (FFA). A follow-up study showed that the angles between the arcade vessels and the optic disk had narrowed (from 110.1° to 99.7° in the right eye, and from 107.1° to 97.4° in the left eye). (**c**) The fundus photograph of the right eye of I:1. The color fundus photography looked normal, and mild FEVR changes were detected using FFA, including the avascular area in the temporal peripheral retina (white arrow), diffused dye leakage (green arrow), and increased straightening of vascular branching. (**d**) The fundus photograph of the right eye of II:2. An extra sign of straightened vessel branching in the peripheral retina was observed (blue arrow). (**e**) The OCT images of the proband with surrogate markers of retinal structure. The proband showed retinoschisis between the inner limiting membrane (ILM) and outer plexiform layer (OPL) of the temporal retina. The red triangle represents the location of the retinoschisis; ILM, inner limiting membrane; NFL, nerve fiber layer; GCL, ganglion cell layer; IPL, inner plexiform layer; INL, inner nuclear layer; OPL, outer plexiform layer; ONL, outer nuclear layer; ELM, external limiting membrane; RPE, retinal pigment epithelium. ①, macula; ②, optic disc.

**Figure 2 genes-14-00587-f002:**
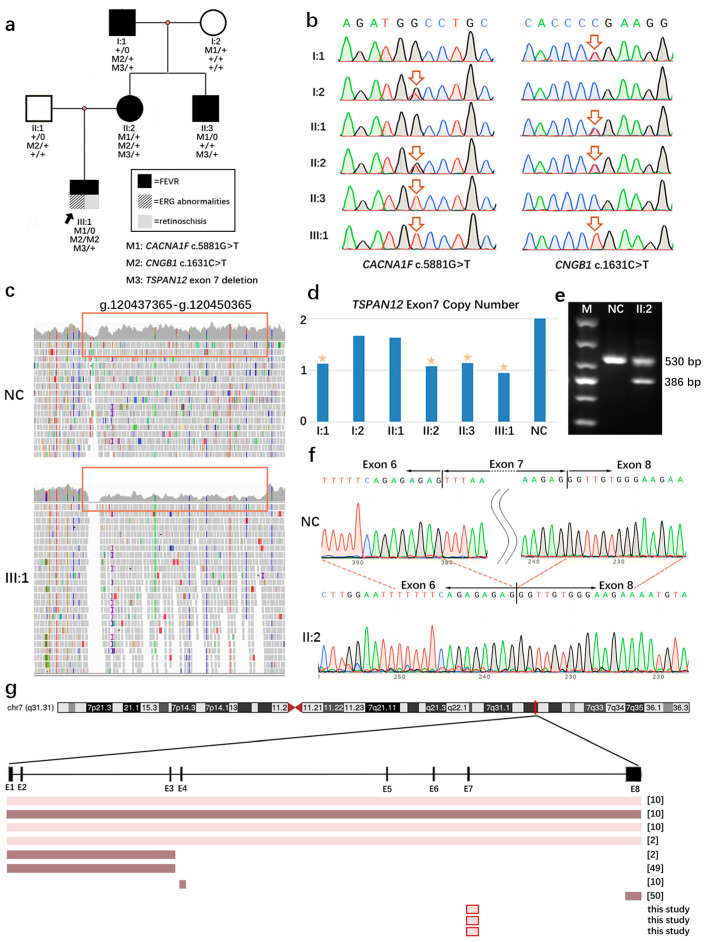
Genetic analysis of the family. (**a**) The pedigree and segregation of detected variants and CNVs in the family. (**b**) *CACNA1F* c.5881G>T and *CNGB1* c.1631C>T variants detected in this family. (**c**) IGV visualization. The gray waveform framed by the orange rectangle represents read coverage. The dwarfed waveform indicates deletion. (**d**) Copy number of exon 7 in *TSPAN12*. A yellow star indicates one copy of exon 7 in *TSPAN12* was detected in I:1, II:2, II:3, and the proband (III:1) using qPCR. (**e**) Gel view of products of the reverse transcription polymerase chain reaction (RT-PCR). Two fragments were observed in II:2, including one fragment of 530 bp and an extra shorter fragment of 386 bp compared to NC (normal control) with only one fragment of 530 bp. The shorter fragment represented the amplicon with the deletion of exon 7 in *TSPAN12*. M, 1000 DNA markers with a brighter band of 400 bp. (**f**) Direct sequence verified the deletion of exon 7 in *TSPAN12* of II:2. (**g**) Schematic representation of deletions in *TSPAN12* encompassing exon regions. Exon regions are highlighted in black, and the connecting line represents intron regions. Bars represent different gross deletions identified in previously reported FEVR patients and this study. The color indicates the FEVR classification in the more severe eye of the patient; light color represents stages 1/2, and dark color represents stages 3/4.

**Figure 3 genes-14-00587-f003:**
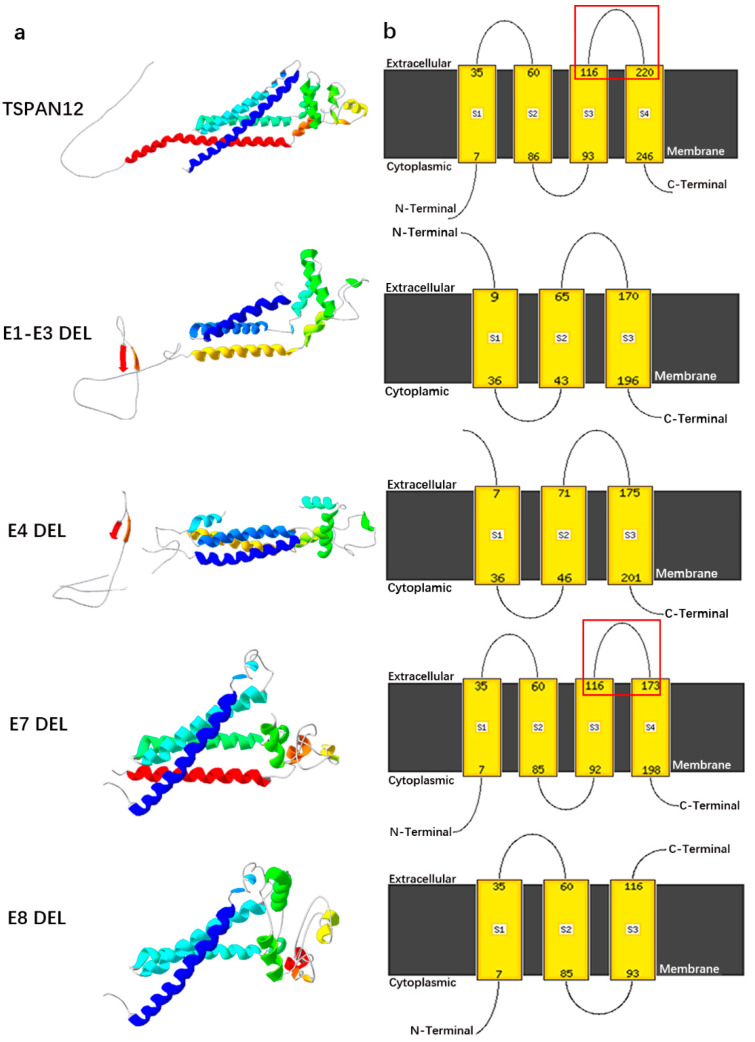
Prediction of 3D structures of TSPAN12 with different deletions. (**a**) Prediction of the 3D structure of TSPAN12 protein after partial exon deletion. (**b**) Prediction of transmembrane helices adopting the topology. The red rectangle frames the large extracellular loop. The ECL-2 region was shrunk when exon 7 was deleted.

## Data Availability

Data supporting the present study are available from the corresponding author upon reasonable request.

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
