# Peer review of "Novel Exon 7 Deletions in TSPAN12 in a Three-Generation FEVR Family: A Case Report and Literature Review"

_genes, 2023, doi:10.3390/genes14030587_

Round 1
Reviewer 1 Report (Previous Reviewer 1)
see attached file

Author Response
Response to Reviewer 1 Comments
Point 1: abstract: ‘cone and rod dystrophy (CORD)’…..see below, as long as you do not have comparative data you cannot state this. You could state ‘ERG abnormalities’
Response 1: Thank you for your helpful advice. This description has already been modified as suggested in the revised manuscript on page 1, line 16.
Point 2: line 159: ‘mother's optometry showed refraction around -2.5 D in both eyes..‘ should read ‘mother's optometry showed refraction was around -2.5 D in both eyes..‘
Response 2: Thank you for your helpful advice. This sentence has already been modified as suggested in the revised manuscript on page 4, line 163.
Point 3: line 161: ‘..Fundus examination revealed a pale optic. should read ‘Fundus examination revealed a pale optic disc.
Response 3: Thanks for your kind advice. We have already revised this word in the revised manuscript on page 4, line 165.
Point 4: line 163-165: were the normal ERG values adjusted for high myopia? if not, please include this in the statement. If normal controls did not show any refractive errors (as you state), then the interpretation of the patient’s ERG responses should be performed with caution.
Response 4: Thanks for your kind advice. We added a comparison of ERG values from the proband with high myopia and pathological myopia to the revised manuscript. This section appeared on page 4, lines 174-180 of the manuscript.
Point 5: 176: ‘result of the fundus photograph‘…should read ‚results of fundus photograph appearance’. 176:
Response 5: Thanks for your kind advice. We have already corrected this sentence as suggested in the revised manuscript on page 4, line 182.
Point 6: Page 7 line 233 and 235 band -> fragment
Response 6: Thanks for your kind advice. We have already corrected this description as suggested in the revised manuscript in line 237, 239, 266-268.
Point 7: line 255: ‘severe abnormalities in the cone and rod mediated responses’: see comment above: as long as you do not have a control group with similar refractive changes you cannot interpret the data. You should weaken you statement such as: ‘….severe abnormalities abnormal cone and rod mediated responses’
Response 7: Thanks for your kind advice. We have already weakened the statement as suggested in the revised manuscript on page 4, line 166, page 9 line 276, and page 11, line 358-362.
Point 8: Figure 2a: CORD should be replaced by ERG abnormalities, see above
Response 8: Thanks for your kind advice. We have already revised this description as suggested in the revised Figure 2a.
Point 9: Please mention and discuss that the exon 7 deletion of TSPAN12 is an in-frame deletion and might lead to a truncated protein. E.g the disrupted ECL-2 of TSPAN12 might interfere with TSPAN12 and FZD4 interaction as suggested by Musada et al. (https://www.ncbi.nlm.nih.gov/pmc/articles/PMC4912735/) or Xiao et al. (https://www.ncbi.nlm.nih.gov/pmc/articles/PMC6885210/)
Response 9: Thanks for your kind advice. The function of ECL-2 and the possible effect of exon 7 deletion were added on page 10, line 316-319, and line 327-332. PMC6885210 was cited as reference 45 and PMC4912735 was cited as reference 48.

Reviewer 2 Report (New Reviewer)
Reviewer’s comments:
The authors studied FEVR family and identify novel deletions in TSPAN12 gene. The ophthalmological and other clinical examination and gene analysis technique which authors applied in this study were regular methods, and adequate to clarifies the information we need. I recommend to accept this case study with subject to major revisions. Following are my specific comments;
In introduction, please mention the prevalence rate of FEVR worldwide and in their population.
I recommend authors to mention the representation of M1, M2 and M3 on the side of figure 2a.
The age of onset of disease is a very critical parameter, I recommend authors to mention it in manuscript as well as in supplementary table.
As exon 7 located in EC2 region of gene, please mention the role of this region, also explain the effect of this due to deletion in discussion section.
I recommend authors to add some already reported functional studies (if any) in discussion in order to provide strong evidence that this in-frame deletion is pathogenic.
Please mention the effect on splicing mechanism due to these variants in discussion section.
I also recommend authors to add 3D protein structure of this gene indicating variant and variant effect on structure. Please make the structure using Swiss-Pdb Viewer. Please mention this in discussion section.
Author Response
Response to Reviewer 2 Comments
Point 1: In the introduction, please mention the prevalence rate of FEVR worldwide and in their population.
Response 1: Thanks for your helpful advice. The incidence of FEVR was added in the revised manuscript on page 1, line 24.
Point 2: I recommend authors mention the representation of M1, M2, and M3 on the side of figure 2a.
Response 2: Thank you for your helpful advice. Figure 2a now includes the representation of M1, M2, and M3 as advised.
Point 3: The age of onset of the disease is a very critical parameter, I recommend authors mention it in the manuscript as well as in the supplementary table.
Response 3: Thank you for your helpful advice. We have already included the mean onset age of FEVR on page 1, line 26-27, described the onset age range of reported patients with CNV changes in TSPAN12 and discussed the relationship between onset age and severity on page 11, line 333-334, and line 345-349.
Point 4: As exon 7 is located in the EC2 region of gene, please mention the role of this region, and also explain the effect of this due to deletion in the discussion section.
Response 4: Thanks for your kind advice. The function of ECL-2 and the possible effect of exon 7 deletion were added on page 10, line 316-319, line 323-332, and line 340-352. We also supplemented the protein structure after exon 7 and ECL-2 deletion respectively in Figure 3.
Point 5: I recommend authors add some already reported functional studies (if any) to the discussion in order to provide strong evidence that this in-frame deletion is pathogenic.
Response 5: Thanks for your kind advice. This part was added in the revised manuscript on page 10, line 316-319, and line 323-332. We also supplemented the protein structure after the exon 7 deletion of TSPAN12 in Figure 3.
Point 6: Please mention the effect on the splicing mechanism due to these variants in the discussion section.
Response 6: Thanks for your kind advice. This part has been added to the revised manuscript on page 10, line 317-319.
Point 7: I also recommend authors add 3D protein structure of this gene indicating variant and variant effects on the structure. Please make the structure using Swiss-Pdb Viewer. Please mention this in the discussion section.
Response 7: Thanks for your kind advice. This part had been added in the revised manuscript on page 4, lines 148-155, page 7, line 243-246, page 11, lines 343-352, and Figure 3.
Round 2
Reviewer 1 Report (Previous Reviewer 1)
The manuscript improved significantly.
Minor comment: Line 232: It seems that the found deletion leads to a simple exon 7 skipping. There is no need for a "new splice site" to receive the observed transcript.
Reviewer 2 Report (New Reviewer)
I approved this manuscript for publication. All concerns have been addressed.
This manuscript is a resubmission of an earlier submission. The following is a list of the peer review reports and author responses from that submission.
Round 1
Reviewer 1 Report
Jiang and Wang describe a novel exon 7 deletion in TSPAN12 in a boy with initially examined due to poor sight and high myopia. The deletion was found by WGS. Further examination of the patient revealed FEVR- related vascular changes. Family member carrying the deletion were asymptomatic however they also showed mild FEVR-related vascular changes when FFA was performed. Based on this description there is still no conclusive evidence that this phenotype is caused by the described deletion.
page 2, line 57: OCT: which system was used? were all available family members examined by OCT?
Page 2 line 58: ERG: please list system, electrodes, dilated/ undilated testing, normative data group (compared with normal controls with high myopia?)
page 3, line 129/ Figure legend 1b: How abnormal was the ERG? Which responses were reduced/ delayed by what %? Was there an electronegative ERG?
Figure 1: ERG: ERG waveforms need to be plotted against control group , otherwise leave it out as a figure
Page 3, line 131: Statement “His parents were asymptomatic with good vision” is contradicted by should by pedigree (figure 2a): the mother is affected? Please clarify. Is she asymptomatic with fundus changes detected by FFA?
page 3, line 128: vision can be tested in this age. The statement might be changed into: was not available or similar
Page 5, line 166: Please show the WGS date (coverage) for this deleted region. Why it is mention “in the region of”? With WGS data, it should be possible to determine the break points exactly.
Page 5, line 168: The authors imply that the found deletion of exon 7 might lead to a loss of function variant. However, deletion of exon 7 in TSPAN12 can lead to an in-frame deletion (exon 7 skipping). Thus, loss of function is not proved and this skipping might result in a gain of function.
Page 5, line 177: I am not convinced about your copy number analysis data by RT-PCR. There is no statistical significance shown. I would suggest performing MLPA or a long range PCR spanning the 13kb to proof the existence of a heterozygous deletion in the patient and its family members.
Page 5, line 183: Please name the genes, which have been analyzed. Have splice variants (deep intronic) and copy number been checked in these genes?
Page 5, line 188: what is the normal change in this angle in myopic children at this age? Was there in increase in myopia in btween age 1y. 9 mo and 4 y 3 mo?
Page 6, line 215: ECL-2, according to UniProt amino acid positions 111-224. Is 69-132 meant as amino acid positions?
Page 7, line 223: This stage is according to which classification system? Pendergast or Kashani?
Please state the transcript numbers for the variants shown in the manuscript.
It should be discussed, if this intra-familial phenotypic variability is due to incomplete penetrance.
Suppl. data were not available.
Author Response
Response to Reviewer 1 Comments
Jiang and Wang describe a novel exon 7 deletion in TSPAN12 in a boy with initially examined due to poor sight and high myopia. The deletion was found by WGS. Further examination of the patient revealed FEVR-related vascular changes. Family members carrying the deletion were asymptomatic however they also showed mild FEVR-related vascular changes when FFA was performed. Based on this description there is still no conclusive evidence that this phenotype is caused by the described deletion.
- page 2, line 57: OCT: which system was used? were all available family members examined by OCT?
Author response: Thanks for your kind suggestion. The OCT scans were acquired using a spectral domain OCT system (software ReVue version 2017.1.0.155, Optovue Inc., Fremont, CA, United States). Other family members did not undergo the OCT examination. We have already added this content as suggested in the revised manuscript of in page 2, line 59-60.
-------------------------------------------------------------------------------------------------------
- Page 2 line 58: ERG: please list system, electrodes, dilated/ undilated testing, normative data group (compared with normal controls with high myopia?)
Author response: Thanks for your kind suggestion. This information was added in the revised manuscript on page 2, line 60-64.
-------------------------------------------------------------------------------------------------------
- page 3, line 129/ Figure legend 1b: How abnormal was the ERG? Which responses were reduced/ delayed by what %? Was there an electronegative ERG?
Author response: Thanks for your kind suggestion. We have modified the images according to your comments in Figure 1e; The results of scotopic ERG and photopic ERG were compared with the normal range of values for this ERG instrument in our hospital. In the scotopic ERG 1). The implicit time of the b-wave of the rod response is normal in both eyes. The amplitude of b-wave the rod response is 56.34μV/46.92μV(OD/OS)(<75μV, severely reduced); 2). The implicit time of the mixed response b-wave was 78 ms/83 ms (reference interval: 57-59 ms, moderately delayed) and the amplitude of the b-wave was 241.6μV/180.7μV(<200~300, moderately reduced); 3) oscillatory potentials had no waveform in both eyes; In the photopic ERG 1)The implicit time of a-wave and b-wave of the cone response was normal in both eyes. The amplitude of the a-wave of the cone response was -31.56μV/-30.79μV(50-124, severely reduced) and the b-wave of the cone response was 25.37μV/31.1μV (109-248μV, severely reduced); 2) The 30 Hz flicker response was severely reduced. There is no electronegative ERG. This part had been added in the revised manuscript on line page 4, line 156-167.
-------------------------------------------------------------------------------------------------------
- Figure 1: ERG: ERG waveforms need to be plotted against control group, otherwise leave it out as a figure
Author response: Thanks for your kind suggestion. We have modified the images according to your comments on Figure 1e on Page 5, line 199.
-------------------------------------------------------------------------------------------------------
- Page 3, line 131: Statement “His parents were asymptomatic with good vision” is contradicted by should by pedigree (figure 2a): the mother is affected? Please clarify. Is she asymptomatic with fundus changes detected by FFA?
Author response: Thanks for your kind suggestion. Our description was inaccurate. We have already corrected the statement as suggested in the revised manuscript.
-------------------------------------------------------------------------------------------------------
- page 3, line 128: vision can be tested in this age. The statement might be changed into: was not available or similar
Author response: Thanks for your kind suggestion. We have already revised the statement as suggested in the revised manuscript.
-------------------------------------------------------------------------------------------------------
- Page 5, line 166: Please show the WGS date (coverage) for this deleted region. Why it is mention “in the region of”? With WGS data, it should be possible to determine the break points exactly.
Author response: Thanks for your kind suggestion. The average depth of WGS was added in the manuscript in page3, line 91. We have added the IGV visualization image (Figure 2c) and the investigation of this deleted region at the transcription level as revised.
- Page 5, line 168: The authors imply that the found deletion of exon 7 might lead to a loss of function variant. However, the deletion of exon 7 in TSPAN12can lead to an in-frame deletion (exon 7 skipping). Thus, loss of function is not proved and this skipping might result in a gain of function.
Author response: Thanks for your kind suggestion. We recollected peripheral blood from the patient's mother, and extracted RNA. Then we performed Sanger sequencing on RT-PCR products, sequencing results indicated that the patient's mother had exon 7 deletion. Just as the reviewer predicted, the deletion of exon 7 in TSPAN12 can lead to an in-frame deletion (exon 7 skipping). We have already added this part as suggested in the revised manuscript.
-------------------------------------------------------------------------------------------------------
- Page 5, line 177: I am not convinced about your copy number analysis data by RT-PCR. There is no statistical significance shown. I would suggest performing MLPA or a long range PCR spanning the 13kb to prove the existence of a heterozygous deletion in the patient and its family members.
Author response: Thanks for your kind suggestion. The method of using qPCR to verify the deletion of large segments of genes has been reported in the literature. We have supplemented the qPCR results in Supplementary Table 3 and added a citation on page 3, line124.
In addition, we supplemented the validation of this deletion at the transcription level. The experimental results validate the existence of this deletion. This part had been added to the revised manuscript.
-------------------------------------------------------------------------------------------------------
- Page 5, line 183: Please name the genes, which have been analyzed. Have splice variants (deep intronic) and copy numbers been checked in these genes?
Author response: Thanks for your kind suggestion. Splice variants (deep intronic) and copy numbers had already been checked in these genes. Related genes have been added to the revised manuscript as required on page 6, line 235-236.
-------------------------------------------------------------------------------------------------------
- Page 5, line 188: what is the normal change in this angle in myopic children at this age? Was there an increase in myopia in between age 1y. 9 mo and 4 y 3 mo?
Author response: Thanks for your kind suggestion. In this paper, we observed changes in this angle in the patient during follow-up, but unfortunately, there is no reference to relevant data in the literature, and we hope to accumulate more similar data in the future. In addition, there wasn’t an increase in myopia between ages 1Y9M and 4Y3M. We have listed visual acuity changes in detail in the revised Supplementary table 1.
-------------------------------------------------------------------------------------------------------
- Page 6, line 215: ECL-2, according to UniProt amino acid positions 111-224. Is 69-132 meant as amino acid positions?
Author response: Thanks for your kind suggestion. The position of amino acids should be 111-224, and we have corrected page 8, line 292.
-------------------------------------------------------------------------------------------------------
- Page 7, line 223: This stage is according to which classification system? Pendergast or Kashani?
Author response: Thanks for your kind suggestion. The stage was classified according to the Kashani classification system. We have already added this information as suggested in the revised manuscript on page 8, line 301-302.
-------------------------------------------------------------------------------------------------------
- Please state the transcript numbers for the variants shown in the manuscript.
Author response: Thanks for your kind suggestion. We have already added the transcript numbers as suggested in the revised manuscript on page 2, line 44.
-------------------------------------------------------------------------------------------------------
- It should be discussed, if this intra-familial phenotypic variability is due to incomplete penetrance.
Author response: Thanks for your kind advice. In this family, we performed FFA tests on all available members with this large deletion (the patient's grandfather and mother), and the results of the FFA yielded a diagnosis of FEVR. (The patient's uncle was allergic to the contrast agent and therefore was not tested). It is worth noting, however, that the degree to which abnormal FFA results affect vision does differ in this FEVR family, and more studies may be needed to explore why such differences exist. Therefore, we do not believe this intra-familial phenotypic variability is due to incomplete penetrance.
-------------------------------------------------------------------------------------------------------
- data were not available.
Author response: Thanks for your kind suggestion. We will re-upload Suppl. data this time.
-------------------------------------------------------------------------------------------------------

Reviewer 2 Report
The authors describe a case of familial exudative vitreoretinopathy and accompanying genetic data.
The manuscript and study will potentially benefit from addressing the following issues:
· It is unclear what the goal of the report is and which clinical parameters and mechanisms have been identified beyond a straightforward description of phenomena.
· A critical review of similar, already published data on TSPAN12 and other relevant clinical and experimental conditions is needed to delineate novel insights and to advance knowledge on the topic.
· A detailed quantitative analysis of structural changes and a systematic comparison to existing data and the eyes of relatives is not included.
· The methods used to generate data (technical and experimental controls) are described insufficiently. For example, structural correlation of FFA appears missing and presentation of OCT data and surrogate markers of retina structure appears inadequate?
· The assessment of confounding factors and differential diagnoses should be expanded and independently validated.
· The information about co-morbidities and of interventions appears insufficient.
· Conclusions and assumptions based on the structural and functional analyses appear premature and require a more detailed description of how the authors arrive at presented conclusions and how they compare to conclusions presented in similar publications and other relevant clinical and experimental literature on familial exudative vitreoretinopathy and related conditions.
· Functional analyses (ERG) appear underdeveloped and their quality needs to be improved.
· Longitudinal data should be included and, at a minimum, comparisons to similar development stages should be provided.
· Conclusions about clinical practice changes are not substantiated adequately with the information provided and the very limited review of the existing literature.
· The manuscript should be proofread carefully to eliminate typographical, grammar and syntax errors.
Author Response
Dec 27, 2022
Dear Editor and Reviewers,
We gratefully thank you for your time spent making constructive remarks and useful suggestions, which have significantly raised the quality of the manuscript and have enabled us to improve the manuscript. Each suggested revision and comment brought forward was accurately incorporated and considered. Below the comments of the reviewers are the response point by point and the revisions are indicated.
Response to Reviewer 2 Comments
The authors describe a case of familial exudative vitreoretinopathy and accompanying genetic data.
The manuscript and study will potentially benefit from addressing the following issues:
- It is unclear what the goal of the report is and which clinical parameters and mechanisms have been identified beyond a straightforward description of phenomena.
Author response: Thanks for your kind suggestion. With this report, we wanted to show that 1) the phenotype of FEVR can be mild and easily misdiagnosed 2) the method of WGS can help improve the diagnosis, and 3) two new phenotypes were found in the proband. These are described in detail in the manuscript
-------------------------------------------------------------------------------------------------------
- A critical review of similar, already published data on TSPAN12and other relevant clinical and experimental conditions is needed to delineate novel insights and to advance knowledge on the topic.
Author response: Thanks for your kind suggestion. We reviewed all the literature which had reported large deletion of TSPAN12 and summarized the location of the deletion and the ocular changes caused by this deletion in Supplementary table 2. However, the sample size is too small, so bias may exist. The interpretation of clinical data and the associated experimental design were improved in the revised manuscript.
-------------------------------------------------------------------------------------------------------
- A detailed quantitative analysis of structural changes and a systematic comparison to existing data and the eyes of relatives are not included.
Author response: Thanks for your kind suggestion. Quantitative analysis of structural changes and a systematic comparison to existing data had been added in the revised manuscript. (Figure 1e, Figure 2, Supplementary table 2)
-------------------------------------------------------------------------------------------------------
- The methods used to generate data (technical and experimental controls) are described insufficiently. For example, the structural correlation of FFA appears missing and the presentation of OCT data and surrogate markers of retina structure appears inadequate?
Author response: Thanks for your kind suggestion. We have already revised as suggested in the revised manuscript including descriptions of retinal markers in the image of OCT, comparisons of patients' ERG with normal control, and validation of this deletion at the transcription level.(Figure1b, e, Figure 2c, d, e)
-------------------------------------------------------------------------------------------------------
- The assessment of confounding factors and differential diagnoses should be expanded and independently validated.
Author response: Thanks for your kind advice. The patient was misdiagnosed as eoHM at the first visit, and deletion was detected with the help of WGS. Then we performed FFA, which is the gold standard for FEVR, on the patient. Combined with the results of the family co-separation, we finally confirmed the diagnosis of this FEVR. We have already added this part as suggested in the revised manuscript on Discussion.
- The information about co-morbidities and interventions appears insufficient.
Author response: Thanks for your kind advice. We have followed this patient for three years. For those with prior evidence, the ophthalmologist in the fundus department recommended continued follow-up with visual acuity exams every six months, and no other intervention for the time being; if retinal vascular traction increases or retinoschisis threatens the macula, further management is required. This part had been added in the revised manuscript on page 6, line 240-243.
-------------------------------------------------------------------------------------------------------
- Conclusions and assumptions based on the structural and functional analyses appear premature and require a more detailed description of how the authors arrive at presented conclusions and how they compare to conclusions presented in similar publications and other relevant clinical and experimental literature on familial exudative vitreoretinopathy and related conditions.
Author response: Thanks for your kind suggestion. We complemented this with functional validation at the transcription level, demonstrating that this reported structural variant results in a truncated CDS, this part had been added in the revised manuscript. All literature related to the large deletion of TSPAN12 was assembled and phenotypes of the patients with these deletions were compared in supplementary table 2.
-------------------------------------------------------------------------------------------------------
- Functional analyses (ERG) appear underdeveloped and their quality needs to be improved.
Author response: Thanks for your kind suggestion. We have modified the images according to your comments in Figure 1e; The results of scotopic ERG and photopic ERG were compared with the normal range of values for this ERG instrument in our hospital. In the scotopic ERG 1). The implicit time of the b-wave of the rod response is normal in both eyes. The amplitude of b-wave the rod response is 56.34μV/46.92μV(OD/OS)(<75μV, severely reduced); 2). The implicit time of the mixed response b-wave was 78 ms/83 ms (reference interval: 57-59 ms, moderately delayed) and the amplitude of the b-wave was 241.6μV/180.7μV(<200~300, moderately reduced); 3) oscillatory potentials had no waveform in both eyes; In the photopic ERG 1)The implicit time of a-wave and b-wave of the cone response was normal in both eyes. The amplitude of the a-wave of the cone response was -31.56μV/-30.79μV(50-124, severely reduced) and the b-wave of the cone response was 25.37μV/31.1μV (109-248μV, severely reduced); 2) The 30 Hz flicker response was severely reduced. There is no electronegative ERG. This part had been added in the revised manuscript on line page 4, line 156-167.
-------------------------------------------------------------------------------------------------------
- Longitudinal data should be included and, at a minimum, comparisons to similar development stages should be provided.
Author response: Thanks for your kind suggestion. On this point, we refer to this literature (PMID: 31294129). This literature had been cited in the article on page 6, line 243.
-------------------------------------------------------------------------------------------------------
- Conclusions about clinical practice changes are not substantiated adequately with the information provided and the very limited review of the existing literature.
Author response: Thanks for your kind suggestion. We have already added this part as suggested in the revised manuscript of This part had been added in the revised manuscript on page 6, line 240-243.
-------------------------------------------------------------------------------------------------------
- The manuscript should be proofread carefully to eliminate typographical, grammar and syntax errors.
Author response: Thanks for your kind suggestion. We had already used the editing services in MDPI. The receipt has been uploaded as an attachment.
-------------------------------------------------------------------------------------------------------

Round 2
Reviewer 1 Report
The authors improved the manuscript by correcting and including the reviewer’s suggestions. However, there are some points, which need to be included/ addressed
Major comments:
Fig. 2 e: I would expect overlapping sequences (due to the heterozygosity of the mother) starting/ending (depending on the primer used for sequencing) at the splice site of exon 6 and 8. The mother should also have a correctly spliced transcript where exon 7 exists. Do you detect two fragments in the RT-PCR of the mother? Which primer pair has been used for fig 2 e?
Please show a gel picture of the RT-PCR products.
Please mention and discuss that exon 7 deletion of TSPAN12 is an in-frame deletion and might lead to a truncated protein. E.g the disrupted ECL-2 of TSPAN12 might interfere with TSPAN12 and FZD4 interaction as suggested by Musada et al. (https://www.ncbi.nlm.nih.gov/pmc/articles/PMC4912735/) or Xiao et al. (https://www.ncbi.nlm.nih.gov/pmc/articles/PMC6885210/)
Some minor comments:
line 42: (NM_012338.4)
line 147: ‘His visual acuity..’ should be corrected: ‘his refraction…’
line 150: ‘mother's optometry showed refraction around -2.5 D in both eyes..‘ should read ‘mother's optometry showed refraction was around -2.5 D in both eyes..‘
line 151: ‘..Fundus examination revealed a pale optic. should read ‘Fundus examination revealed a pale optic disc.
line 152: ‘…in the cone and rod cell responses. ‘ should be better ‘…in the cone and rod cell responses OR in the cone and rod mediated responses’
line 154: were the normal ERG values adjusted for high myopia? if not, please include this in the statement
line 154 ff: thank you for providing the detailed ERG data.
line 159: No oszillatory potentials could be measured.
Fig 1e; The label Ops (better OPs) and Max response are mixed up
Fig 1 e: Are the plotted ERG control data from a subject with the same refractive error? if not, please state the refraction of the control in the figure legend
Line 205-207: This section still implies that the found variant leads to a loss of function. Which probably is not true.
line 211: acceptor
line 213: you could add ‘asymptomatic’ to family members
Author Response
Response to Reviewer 1 Comments
Point 1: Fig. 2 e: I would expect overlapping sequences (due to the heterozygosity of the mother) starting/ending (depending on the primer used for sequencing) at the splice site of exon 6 and 8. The mother should also have a correctly spliced transcript where exon 7 exists. Do you detect two fragments in the RT-PCR of the mother? Which primer pair has been used for fig 2 e?
Response 1: Thanks for your kind advice. We had already found two fragments in the mother's RT-PCR products, and the gel image and detailed descriptions of the RT-PCR products had been added to Figure 2e and figure legend. TSPAN12-E5/6-F1 5'- AGGAACTTATGGTTCCAGTACAA-'3 and TSPAN12-E8-R1 5'- TGCCATGGATGTGTTCAA-'3 were the primers we used. The detailed primer pair description has been added to the Method on page 4, lines 145-148.
Point 2: Please show a gel picture of the RT-PCR products.
Response 2: Thanks for your kind advice. Figure 2e now includes a gel image of the RT-PCR products. The agarose gel electrophoresis result was described in the DISCUSSION on page 7, line 230-236.
Point 3: Please mention and discuss that the exon 7 deletion of TSPAN12 is an in-frame deletion and might lead to a truncated protein. E.g the disrupted ECL-2 of TSPAN12 might interfere with TSPAN12 and FZD4 interaction as suggested by Musada et al. (https://www.ncbi.nlm.nih.gov/pmc/articles/PMC4912735/) or Xiao et al. (https://www.ncbi.nlm.nih.gov/pmc/articles/PMC6885210/)
Response 3: Thanks for your helpful advice. This section of the discussion has been added to the revised manuscript on page 7, lines 228-236, and lines 260-263.
Point 4:
line 42: (NM_012338.4)
line 147: ‘His visual acuity..’ should be corrected: ‘his refraction…’
line 150: ‘mother's optometry showed refraction around -2.5 D in both eyes..‘ should read ‘mother's optometry showed refraction was around -2.5 D in both eyes..‘
line 151: ‘.Fundus examination revealed a pale optic. should read ‘Fundus examination revealed a pale optic disc.
line 152: ‘…in the cone and rod cell responses. ‘ should be better ‘…in the cone and rod cell responses OR in the cone and rod mediated responses’
Response 4: Thanks for your kind advice. We have already revised them one by one as suggested in the revised manuscript.
Point 5:
line 154: were the normal ERG values adjusted for high myopia? if not, please include this in the statement
line 154 ff: thank you for providing the detailed ERG data.
Response 5: Thanks for your kind advice. Normal ERG values were not corrected for myopia. This section was added on page 4, line 161-174 of the revised manuscript.
Point 6: line 159: No oscillatory potentials could be measured.
Response 6: Thanks for your kind advice. The description of oscillatory potentials has been removed from the revised manuscript.
Point 7: Fig 1e; The label Ops (better OPs) and Max response are mixed up
Response 7: Thanks for your kind advice. We mislabeled these two images and have corrected them, as you pointed out.
Point 8: Fig 1 e: Are the plotted ERG control data from a subject with the same refractive error? if not, please state the refraction of the control in the figure legend
Response 8: Thanks for your kind advice. There are no significant ocular abnormalities or refractive errors in the ERG control data of NC (normal control). This description has been added to Figure 1d's legend.
Point 9: Line 205-207: This section still implies that the found variant leads to a loss of function. Which probably is not true.
Response 9: Thanks for your kind advice. This section of the statement has been corrected in the revised manuscript on page 7, line 223-226 and line 249-251.
Point 10: line 211: acceptor
Response 10: Thanks for your kind advice. The acceptor's description was incorrect; we have corrected this in the revised manuscript.
Point 11: line 213: you could add ‘asymptomatic’ to family members
Response 11: Thanks for your kind advice. This has been revised as suggested.

Reviewer 2 Report
Major concerns remain after the revision:
· The stated goal of the report was not accomplished with results based on a single case.
· A quantitative analysis of structural changes and a systematic comparison to existing data and the eyes of relatives is still not included despite the assertion of the authors to the contrary in the rebuttal letter.
· The methods used to generate data (technical and experimental controls) are described insufficiently. For example, structural correlation of FFA appears missing and presentation of OCT data and surrogate markers of retina structure appears inadequate?
· The assessment of confounding factors and differential diagnoses remains rudimentary and should be independently validated.
· The information about co-morbidities and of interventions appears still insufficient.
· The evaluation and quantitative assessment of the functional tests (ERG) appear still underdeveloped.
· Longitudinal data for the single case should be included. As this case represents a development stage, an individual point in time without adequate follow-up appears insufficient to draw meaningful conclusions.
Author Response
Response to Reviewer 2 Comments
Point 1: The stated goal of the report was not accomplished with results based on a single case.
Response 1: Thanks for your helpful advice. A single example is insufficient to demonstrate our initial point. We have revised the title of the article to report this case more critically. Other enhanced statements can be found on page 1, line 15-16, and page 2, line 43-45.
Point 2: A quantitative analysis of structural changes and a systematic comparison to existing data and the eyes of relatives are still not included despite the assertion of the authors to the contrary in the rebuttal letter.
Response 2: Thank you for your helpful advice. In DISCUSSION, we compared phenotypes caused by large deletions of TSPAN12, and the severity of the phenotype was not related to the size of the deletion but to its location (Figure 2g). The ocular phenotype and severity of the family are described on page 6, lines 208-213, and page 7, lines 243-247.
Supplementary Table 1 now includes the differences in ocular performance between family members and the proband.
Point 3: The methods used to generate data (technical and experimental controls) are described insufficiently. For example, the structural correlation of FFA appears missing and presentation of OCT data and surrogate markers of retina structure appears inadequate?
Response 3: Thank you for your helpful advice. In terms of the FFA structure's relevance, we used the angular variation of the vascular arcade (the angles are between the arcade vessels and centering on the optic disks) to represent the disease state as reported in the literature. The linearization of the vascular arcade in this case strongly suggests disease progression.
Furthermore, we changed Figure 1 by replacing the OCT images with clearer ones and marking the surrogate markers of retina structure (Supplemental Table 1), adding more visual FFA images and descriptions of relatives (Supplemental Table 1), and changing the figure legend on page 6, line 214-220.
Point 4: The assessment of confounding factors and differential diagnoses remains rudimentary and should be independently validated.
Response 4: Thanks for your kind advice. This part had been added in the revised manuscript on page 7, line 253-266.
Point 5: The information about co-morbidities and of interventions appears still insufficient
Response 5: Thanks for your kind advice. Previously literature showed that among asymptomatic mild FEVR patients, although only 1.61% (2 out of 124 eyes) would develop into severe situations that required laser photocoagulation treatment, all patients with mild FEVR need lifelong monitoring. Follow-up examinations were performed on the patient in this study, beginning with his first visit to our hospital and continuing for approximately six months. Developing retinal vascular traction and enlarging retinoschisis were observed, but further surgery or laser treatment measures were not taken by now considering his age and macular non-involvement after multidisciplinary treatment. This part had been added to the revised manuscript on page 8, line 269-284.
Point 6: The evaluation and quantitative assessment of the functional tests (ERG) appear still underdeveloped
Response 6: Thanks for your kind advice. We have added these contents following your suggestions. The percentages of each ERG response's degree of decline/delay have been supplemented on page 4, line 161-174.
Point 7: Longitudinal data for the single case should be included. As this case represents a development stage, an individual point in time without adequate follow-up appears insufficient to draw meaningful conclusions.
Response 7: Thanks for your kind advice. We are still following up with this patient. Changes in the patient's OCT, FFA, and visual acuity during the follow-up have been supplemented. This part had been added in the revised manuscript on page 8, line 272-284, as well as Supplementary Table 1.
